# Sex-Dependent Effect of Chronic Piromelatine Treatment on Prenatal Stress-Induced Memory Deficits in Rats

**DOI:** 10.3390/ijms24021271

**Published:** 2023-01-09

**Authors:** Natasha Ivanova, Milena Atanasova, Zlatina Nenchovska, Jana Tchekalarova

**Affiliations:** 1Institute of Neurobiology, Bulgarian Academy of Sciences (BAS), 1113 Sofia, Bulgaria; 2Department of Biology, Medical University of Pleven, 5800 Pleven, Bulgaria

**Keywords:** prenatal stress, piromelatine, memory, melatonin, pCREB/CREB

## Abstract

Prenatal stress impairs cognitive function in rats, while Piromelatine treatment corrects memory decline in male rats with chronic mild stress. In the present study, we aimed to evaluate the effect of chronic treatment with the melatonin analogue Piromelatine on the associative and spatial hippocampus-dependent memory of male and female offspring with a history of prenatal stress (PNS). We report that male and female young adult offspring with PNS treated with a vehicle had reduced memory responses in an object recognition test (ORT). However, the cognitive performance in the radial arm maze test (RAM) was worsened only in the male offspring. The 32-day treatment with Piromelatine (20 mg/kg, i.p.) of male and female offspring with PNS attenuated the impaired responses in the ORT task. Furthermore, the melatonin analogue corrected the disturbed spatial memory in the male offspring. While the ratio of phosphorylated and nonphosphorylated adenosine monophosphate response element binding protein (pCREB/CREB) was reduced in the two sexes with PNS and treated with a vehicle, the melatonin analogue elevated the ratio of these signaling molecules in the hippocampus of the male rats only. Our results suggest that Piromelatine exerts a beneficial effect on PNS-induced spatial memory impairment in a sex-dependent manner that might be mediated via the pCREB/CREB pathway.

## 1. Introduction

Constant stress on an expectant mother can influence the brain development of the embryo and the fetus [1,2]. Animal and human studies have shown that chronic adverse events through gestation decrease brain grey matter and increase stress hormones during and beyond the pregnancy, which brings about lifelong neurodevelopmental consequences [3,4,5,6,7]. The latter also includes a lower intellectual state and a decline in memory and cognitive performance [3,8,9]. In contrast, other investigators report that prenatal stress can have a dissimilar, advantageous or disadvantageous male–female memory impact [10,11].

Brain-derived neurotrophic factor (BDNF) is involved in neuronal plasticity, hippocampal synaptogenesis and spatial memory formation [12,13,14]. Deterioration of BDNF is associated with depression, disturbed circadian rhythms and cognitive deficits [1,13,15]. This essential for brain plasticity molecules, via binding to the receptor tyrosine kinase B (TrkB), activate several signaling pathways such as the Ras-mitogen-activated protein kinase (MAPK), the phosphatidylinositol-3-kinase (PI3K) and the phospholipase Cγ (PLC-γ) pathway [16]. BDNF can affect gene regulation related to long-term potentiation (LTP) via MARK-cyclic adenosine monophosphate response element binding protein (CREB) signaling [16]. Alteration of CREB negatively impacts long-term reminiscence and associative and spatial memory [17,18,19]. The link between BDNF and CREB is bi-directional. While as a transcription factor CREB controls BDNF expression, neurotrophine triggers CREB phosphorylation [20,21]. Moreover, the literature data suggest a sexually dichotomous influence of prenatal stress on CREB and BDNF expression [11,15,22]. In addition, CREB activation could be elevated through serotonin (5-HT) activation of the second-messenger cAMP-PKA [21].

The literature data suggest that the melatoninergic system influences the memory and learning processes [23,24,25] via CREB signaling [26,27]. While melatonin modifies CREB phosphorylation [26], the latter is associated with enhanced melatonin synthesis [27]. 

The novel compound Piromelatine (Pir), N-(2-[5-methoxy-1H-indol-3-yl]ethyl)-4-oxo-4Hpyran-2-carboxamide, is a multimodal drug with a complex mechanism of action, which simultaneously targets the melatoninergic and, in part, the serotoninergic systems by activating the MT type 1, 2 and 3 receptors and the serotonin (5-HT) type 1A and 1D receptors [28]. A phase-II clinical study has indicated a good capacity of the drug for insomnia treatment [28]. Studies, including our recent ones, demonstrate that Pir improves the negative changes induced by prenatal stress on emotional behavior, disturbed circadian locomotion and sleep/wake cycle rhythm and neuro- and the biochemical profile of offspring with a history of prenatal stress (PNS) in a sex-dependent manner [6,7,15,29,30]. Additionally, preclinical reports suggest that this drug prevents cell loss and exerts a beneficial action on Aβ42-induced cognitive impairment in a rat model of Alzheimer’s disease (AD) [31]. Recently, we reported that Piromelatine augmented BDNF expression in the hippocampus of male and female offspring with a history of prenatal stress [15]. This novel compound also targets the 5-HT type 1A receptors, which are widely expressed in the brain sections involved in the learning and memory processes [32,33].

With all these findings in mind, we assumed that the multimodal compound Pir can favorably affect cognitive and memory disturbances in the offspring of mothers stressed during pregnancy. Being an activator of both the melatonin and the 5-HT receptors, affirmatory to the memory and cognition exploration, we investigated the effects of Pir on melatonin and CREB in the hippocampus.

## 2. Results

### 2.1. The Chronic Piromelatine Treatment Corrected Impairments of Associative Memory in Male and Female Offspring with a History of Prenatal Stress

The effect of the drug Piromelatine on associative memory was assessed in the ORT in male and female rats. The time and frequency (counts) that each rat explored on a novel and familiar object were used to calculate the discrimination index. Male adult offspring with a history of prenatal stress demonstrated a lack of preference for to the novel object (PNS-veh males: time: *p* < 0.001 vs. C-veh (Figure 1A) and counts: *p* = 0.004 vs. C-veh (Figure 1B)). While chronic Piromelatine treatment in male control rats did not change the response in the ORT task (*p* > 0.05 vs. C-veh group), the drug reduced the prenatal stress-induced memory deficit in male rats (PNS-Pir males: time: *p* = 0.019 vs. PNS-veh group (Figure 1A) and counts: *p* = 0.04 vs. C-veh (Figure 1B)).

Like male offspring, prenatally-stressed female rats treated with a vehicle exhibited a lower preference to explore the novel object compared to controls treated with vehicle (PNS-veh females: time: *p* = 0.022 (Figure 1C) and counts: *p* = 0.036 vs. C-veh (Figure 1D)). There was no difference in the discrimination index (time and counts) between the C-veh and C-Pir female groups (*p* > 0.05) (Figure 1C,D). However, chronic Piromelatine treatment mitigated the impaired response in the ORT of rats with PNS (time: *p* = 0.0144 vs. PNS-veh group) (Figure 1C).

### 2.2. The Chronic Piromelatine Treatment Corrected Impairments of Hippocampus-Dependent Spatial Memory in Male Offspring with a History of Prenatal Stress

The effect of Piromelatine on hippocampus-dependent spatial memory was evaluated in a RAM test. Like the controls treated with a vehicle, the Piromelatine treatment showed a tendency to improve the performance with repeated sessions. Thus, the control group with Piromelatine had lower WMEs (*p* = 0.05 and *p* = 0.045 in the fourth and fifth session compared to the first session, respectively), DWMEs (*p* = 0.05 in the fifth session compared to the first session) and time (*p* = 0.041 the fifth session compared to the first session) (Figure 2A–C). The male adult offspring with prenatal stress did not succeed in improving their performance over time, and there was no significant difference among the five sessions with respect to the WMEs (*p* > 0.05) (Figure 2A), DWMEs (*p* > 0.05) (Figure 2B) and the time to complete the task (*p* > 0.05) (Figure 2C). The Piromelatine group with a history of prenatal stress exhibited improved responses in the last sessions with diminished WMEs (*p* = 0.043 and *p* = 0.05 in the fourth and fifth session compared to the first session, respectively), DWMEs (*p* = 0.05 the fifth session compared to the first session) and time (*p* = 0.043 the fifth session compared to the first session) (Figure 2A–C).

Surprisingly, the female adult offspring with a history of prenatal stress showed lower WMEs (*p* = 0.0307 and *p* = 0.0037 in the fourth and fifth session compared to the first session, respectively), DWMEs (*p* = 0.019 in the fifth session compared to the first session) and time (*p* = 0.0059 the fifth session compared to the first session) (Figure 2D–F). The female control group treated with Piromelatine showed similar results to the control vehicle group’s performance during the five sessions of the RAM task. Moreover, the C-veh group showed significantly fewer WMEs in the fifth session vs. the first session (*p* = 0.043) and a decreased time to complete the task (*p* < 0.05 the fourth and fifth session compared to the first session). The Piromelatine treatment, in the prenatally stressed rats, demonstrated improved responses in the RAM task from the second to the last session compared to the first session (WMEs: *p* < 0.05 vs. first session; DWMEs: *p* < 0.05 vs. first session) (Figure 2D,E).

### 2.3. The Chronic Piromelatine Treatment Tended to Increase Melatonin Levels in Male Rats with a History of Prenatal Stress 

The effect of the melatonin MT1/MT2 receptor agonist Piromelatine on the melatonin plasma level was studied in the male and female offspring with prenatal stress. Prenatal stress did not affect the plasma melatonin levels in the male and female rats (*p* > 0.05 PNS-veh vs. C-veh group) (Table 1). The Piromelatine treatment in the male offspring with PNS tended to elevate the release of melatonin in plasma compared to the offspring treated with a vehicle without reaching a significant difference (*p* > 0.05). This melatonin analogue did not affect hormonal release in the female rats with a history of prenatal stress (*p* > 0.05).

### 2.4. The Chronic Piromelatine Treatment Exerted a Sex-Dependent Elevation of Phosphorylated and Nonphosphorylated Adenosine Monophosphate Response Element Binding Protein (pCREB/CREB) Ratio in the Hippocampus in a PNS Model

The PNS exposure suppressed the phosphorylation of CREB, and the pCREB/CREB ratio was significantly reduced both in the male PNS-veh group (*p* = 0.0012 vs. C-veh group) (Figure 3A) and in the female PNS-veh group (*p* = 0.002 vs. C-veh) (Figure 3B). In the Piromelatine-treated controls, the drug did not produce any changes in the ratio of the neuroplasticity markers in the hippocampus compared to the controls treated with a vehicle (Figure 3A,B). This melatonin analogue was able to correct the PNS-induced decrease in the pCREB/CREB ratio in the male group (*p* = 0.0005 vs. PNS-veh group) but was ineffective in the female rats (*p* = 0.0364 PNS-Pir vs. C-veh).

## 3. Discussion

The present findings revealed that Piromelatine corrected the declined spatial memory in a sex-dependent manner in offspring with a history of PNS via the pCREB/CREB signaling pathway in the hippocampus.

Our results suggested that Piromelatine, administered for about 32 days, did not possess an adverse effect on short-term associative memory in the ORT test and the hippocampus-dependent spatial memory in the RAM test in the male and female control offspring. The drug was chronically injected intraperitoneally at a dose of 20 mg/kg (i.p.) that did not affect locomotor activity [15], thereby eliminating possible false positive responses. Our results agreed with the report of He et al. (2013) [31], who demonstrated that a single administration of Piromelatine at a dose of 50 mg/kg either in the morning or in the afternoon did not change both the short- and long-term memory in an ORT test of male Sprague–Dawley rats. Recently, Cattaneo et al. (2019) [34] also reported that male and female offspring with a history of prenatal restrain stress in the last week of pregnancy demonstrated a decreased discrimination index in an ORT test. In the present work, pregnant rats were exposed to variable stressors during the second and last week of pregnancy (E7 to P0), suggesting that the mid-gestational period is also a crucial period for epigenetic plastic changes affecting cognitive function in the two sexes. In addition, Cattaneo et al. (2019) [34] demonstrated that a disrupted response in an ORT test of male and female PNS rats must be related to the changed transcription of important genes associated with the plastic reorganization, inflammation, protein kinase A and glucocorticoid signaling pathway in the dorsal hippocampus. The data we reported earlier, with the same PNS protocol, also confirmed the essential role of glucocorticoid receptors as well as a disrupted feedback mechanism of hypothalamic–pituitary–adrenal (HPA) axis in the impairment of the behavioral functions in male and female offspring [6]. Piromelatine alleviated the PNS-induced impaired ORT task both in the male and in female rats, suggesting that this melatonin analogue has a potential cognitive-facilitating activity in different rat models that are associated with memory impairment. Recently, He et al. (2013) and Fu et al. (2016) also reported that treatment with this melatonin analogue improved cognitive responses in ORT tasks in two other models characterized by memory decline, an intrahippocampal Aβ(1–42)-induced AD model and a chronic mild stress model of depression, respectively [31,35]. Moreover, unlike melatonin, Piromelatine was found to have a more substantial effect that was not dependent on the time of injection (i.e., in the morning vs the afternoon), suggesting an essential role of its affinity to 5-HT1A/1D receptors in the cognitive-potentiating activity of this drug [31].

Unlike the impaired associative memory in the ORT test, a sex-related difference in the hippocampus-dependent spatial memory, tested in the RAM test, was found in the present study. This result supports the hypothesis about sex divergence in PNS-induced consequences on cognitive abilities. Indeed, Mueller and Bale (2007) [36] found a critical gestational time window for memory changes in PNS male and female offspring. While female offspring rats with early PNS were found to have better performance in the memory task compared to their matched controls, male offspring showed worse memory responses than controls [36]. In addition, Zuena et al. (2008) [11] revealed that PNS conducted in the mid and late gestational period, as applied in our study, impaired the spatial memory responses of males but improved those of female rats in the Morris water maze task. However, these authors reported that sex-dependent PNS-induced changes in the hippocampus-dependent spatial memory were not associated with the plastic signaling molecule brain-derived neurotrophic factor (BDNF) expression in the hippocampus. In contrast, recently we reported that BDNF expression was decreased in male offspring with PNS but was not affected in female offspring [15], suggesting that this signaling molecule in the hippocampus is critical for improved responses in the RAM task of female rats with PNS. We cannot exclude a possible effect of the estrous cycle on spatial memory in female rats. This is a limitation, which we were not able to avoid, because the estrous cycle is very short (4–5 days), and RAM test execution requires at least two weeks. In line with our findings, Shang (2019) [37] reported that spatial memory impairment in the water maze task of male rats with a history of restrained PNS in the late gestational period was associated with diminished BDNF/TrkB signaling and reduced pCREB/CREB ratio. The literature data support the presumption about a bi-directional positive link between the two plastic molecules involved in cognitive functions. While CREB induces BDNF expression through binding to the gene-promotor area [20], the neurotrophic factor triggers the phosphorylation of CREB [21]. Furthermore, we reported earlier that Piromelatine enhanced the BDNF level in the hippocampus of the two offspring sexes with a history of prenatal stress [15]. The latter corresponds with the present findings that Piromelatine exerts sex-dependent changes in the pCREB/CREB ratio. In addition, considering the data of Maronde et al. (1997) [27] regarding the increased synthesis of melatonin following CREB phosphorylation in rat pinealocytes, we can speculate that Piromelatine treatment mediates pCREB-induced melatonin synthesis in PNS male offspring. On the other hand, it is plausible to assume that this agonist of MT1/MT2 receptors regulates pCREB/CREB signaling in a pathological state only in male rats, but not in physiological conditions with an intact melatonin system.

## 4. Materials and Methods

### 4.1. Animals

Standard housed (light–dark cycle (12/12); temperature: 21 ± 1 °C; humidity: 50–60%; animals per cage: 3–4) male and female rats of the Sprague–Dawley strain (Charles River, Italy), with body weights approximately 220–250 g, supplied with food and water ad libitum, were used for the experimental design. The treatment of animals was carried out according to the Declaration of Helsinki Guiding Principles on Care and Use of Animals (DHEW Publication, NHI 80-23) and the European Communities Council Directives of 24 November 1986 (86/609/EEC). The project was approved by the Bulgarian Food Safety Agency.

### 4.2. Prenatal Stress Procedure

Following a week of adaptation, adult female rats were paired with males for breeding. The presence of a copulatory plug was affirmed as gestation day 0 (E0). Apart from the controls, half of the pregnant rats were moved to a different room and exposed to a stress procedure as previously described [6]. In brief, different types of random stressors to avoid habituation (one short-term during the day and one long-term overnight) were applied daily starting from day E7 until birth (postnatal day (P) 0), while controls were left undisturbed. At weaning (P21), 9 litters with no less than 8 pups with a similar sex ratio were exploited. Offsprings of at least 3 dams were used per group, randomly treated with vehicle (veh) or Pir. The experimental (PNS) and control (C) groups from both sexes consisted of sexually mature offspring rats (P60) for the behavioral tests and biochemical analyses (*n* = 8).

### 4.3. Drug Administration and Protocol Design

Piromelatine (kindly gifted by Neurim Pharmaceuticals Ltd.), dissolved in hydroxyethyl cellulose 1%, was administered i.p. (20 mg/kg) at 4:00 p.m. (two hours before the onset of the dark phase), starting from P60 for about 32 days until the rats were sacrificed (P92). The dose and the time of application were determined as previously described [6]. The controls and their matched PNS groups received vehicles. Eight groups were assigned to the experimental protocol: C-veh: male and female rats, respectively; C-Pir: male and female rats, respectively; PNS-veh: male and female rats, respectively; and PNS-Pir: male and female rats, respectively.

### 4.4. Behavioral Tests

The conduction of the behavioral tests started from P74 (after the 13th injection of veh/Pir) and was between 10:00 a.m. and 2:00 p.m. The tests were conducted by a blinded researcher in a sound-protected room with diffused light where the rats were moved 30 min before the test procedure. The ORT was conducted at P74 (habituation) and P75 (the test). The RAM test was conducted on P83–P89.

#### 4.4.1. Object Recognition Test (ORT)

Associative memory was detected by an ORT. The ORT apparatus consisted of a black box (50 cm × 50 cm) of polystyrene Plexiglas. The testing was performed on 2 consecutive days: habituation on the first day for 10 min, and training and test sessions on the second day. Twenty-four hours after habituation, a training session (pre-test) and a test session were executed 60 min later for 5 min. In the training session, the rats explored 2 identical objects, while in the test session one of the training objects was replaced with a novel one. The rats which showed a preference to one of the identical objects during the training session were excluded from the statistics. Familiar and novel object exploration counts and time (i.e., sniffing or brief touching) were manually detected by an experienced researcher. The following measures were exploited to determine novel object preference: (1) discrimination index counts sniffing and (2) discrimination index time shifting (seconds) by using the following formula: time or counts novel object/time or counts novel object + time or counts familiar object × 100. The apparatus was cleaned with an acetic acid solution (0.1%) after each tested rat.

#### 4.4.2. Radial Arm Maze

Spatial learning and memory were evaluated by an 8-arm radial maze (RAM) (Harvard Biosci, Holliston, MA, USA). The stainless-steel RAM apparatus consisted of eight identical arms (42 × 12 × 12) radiating from a central octagonal platform (30 cm in diameter) elevated 50 cm off the floor. For the purpose of facilitating spatial navigation, picture signals (circle, square, triangle and star) were available around the apparatus. A week before the training and the test, rats were put on a diet for at least 15% of their body weight. Before the test performance, rats were habituated for 3 days by exploring the maze for 15 min per day with food pellets in each of the eight arms (shaping). After the habituation, the animals performed the RAM task: 5 trials were conducted in total with one session per day for 5 consecutive days. A food pellet was placed in each arm, and the session ended when all baits were found or after 10 min passed. The decrease in the number of errors for each trial was assessed. The second entry into an arm with an already-eaten food pellet was considered a working memory error, while the third entry into an arm with an already retrieved food pellet was calculated as a double working memory error. The following measures were calculated to evaluate the memory achievement: (1) total time to complete the session, (2) the number of working memory errors and (3) double working memory errors. After each tested animal, the apparatus was cleaned with 0.1% acetic acid solution.

### 4.5. Biochemical Methods

The rats were sacrificed 3 days after the last behavioral test (RAM) on P92 2 h before the onset of the dark phase (4 p.m.) under light anesthesia with CO_2_. Trunk blood was collected and both hippocampi were isolated and dissected.

#### 4.5.1. Measurement of Melatonin Levels in the Plasma 

The trunk blood was collected in vacutainer blood collection tubes and kept on ice until centrifuged at 4000 rpm for 10 min. The plasma was stored at −20 °C until assayed. Plasma melatonin levels were measured by an Elisa Plate Reader InfiniteF200Pro, TECAN, Austria, and an ELISA kit (Enzo, Switzerland, cat. No ENZ-KIT150-0001) utilizing the instructions of the manufacturer. Duplicative measurement with mean calculation was applied for each sample, and the results were expressed as ng/mL.

#### 4.5.2. Measurement of CREB1 and Phosphorylated CREB in the Hippocampus

The isolated hippocampi were kept on ice, weighed and preserved at −20 °C until homogenization in cold PBS buffer (pH 7.4) containing 1 mM EGTA, 50 mM NaF, 1 mM EDTA and 1 mM PMSF. After centrifugation of the tissue homogenate at 11,000× *g*, 4 °C for 10 min, CREB1 was measured in duplicates using an ELISA kit (Elabscience cat. No E-EL-R0289)) in pmol/mL. The phosphorylated CREB was measured in duplicates by an ELISA kit of Sunlong, cat. No SL1344Ra, and the concentration was expressed as pg/mL.

### 4.6. Statistical Analysis

All results were presented as mean ± SEM. SigmaStat^®^ (version 11.0., San Jose, CA, USA) and GraphPad Prism 6 software were used for statistical analyses. Experimental data were evaluated by three-way ANOVA for the behavioral tests: RAM with the factors treatment (PNS and drug), sex and trial, and ORT with the factors PNS, sex and drug; for CREB and pCREB with the factors PNS, sex and drug and two-way ANOVA for melatonin with the factors PNS and drug. Post hoc comparisons between groups were conducted by Bonferroni’s t-test or the Mann–Whitney U-test. A *p*-value < 0.05 was accepted as indicating a statistically significant difference.

## 5. Conclusions

The melatonin analogue Piromelatine exerted a beneficial effect on the PNS-induced disturbed associative memory in both the male and female offspring. The hippocampus-dependent impairment of spatial memory was corrected by the drug in the male rats with PNS via the pCREB/CREB signaling pathway. Our results confirmed the potential beneficial effects of this melatonin analogue in pathological conditions related to memory impairment in a sex-dependent manner.

## Figures and Tables

**Figure 1 ijms-24-01271-f001:**
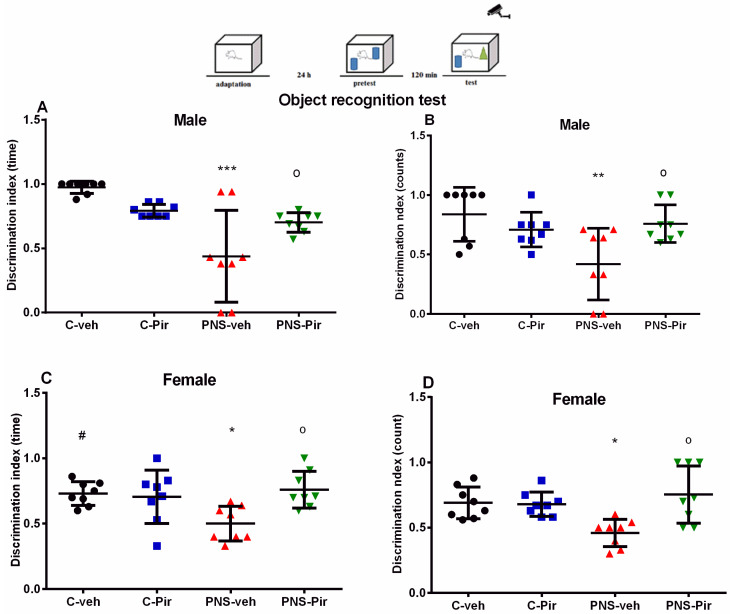
The chronic Piromelatine treatment exerted a beneficial effect on cognitive impairment in male offspring with a history of prenatal stress in the ORT test. Three-way ANOVA indicated a main effect of PNS [F_1,54_ = 11.374, *p* < 0.001], PNS × Drug interaction [F_1,54_ = 15.971, *p* < 0.001] and PNS × sex interaction [F_1,54_ = 5.736, *p* < 0.021] on discrimination index (DI) (time) (**A**,**C**) and PNS × drug interaction [F_1,54_ = 12.929, *p* < 0.001] on DI (counts) (**B**,**D**). Data are presented as means ± SEM: * *p* < 0.005 vs. C-veh, ^o^ *p* < 0.005 vs. PNS-veh, # *p* < 0.05 vs. male.

**Figure 2 ijms-24-01271-f002:**
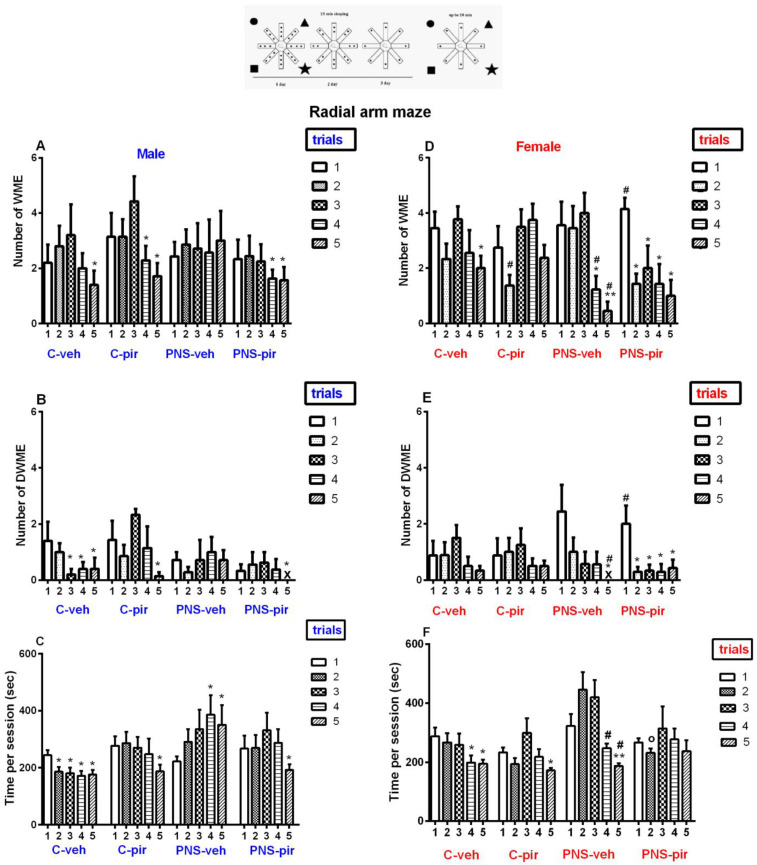
The chronic Piromelatine treatment exerted a beneficial effect on cognitive impairment of offspring with a history of prenatal stress in RAM test. Three-way ANOVA indicated a main treatment effect [F_3,296_ = 2.672, *p* = 0.047], a main trial effect [F_4,296_ = 6.370, *p* < 0.001] and interaction between treatment × sex × trial [F12,296 = 2.98, *p* = 0.05] on the number of WMEs (**A**,**D**), a main trial effect [F_4,296_ = 5.596, *p* < 0.001] and interaction between treatment × sex × trial [F_12,296_ = 1.917, *p* = 0.033] on the number of DWMEs (**B**,**E**), a main treatment effect [F_3,296_= 15.100, *p* < 0.001], a main trial effect [F_3,296_ = 5.594, *p* < 0.001] and interaction between treatment × sex × trial [F_12,296_ = 2.559, *p* = 0.003] on time per session (**C**,**F**). Data are presented as means ± SEM. * *p* < 0.05 vs. first session, ^o^ *p* < 0.05 vs. PNS-veh, # *p* < 0.05 vs. male.

**Figure 3 ijms-24-01271-f003:**
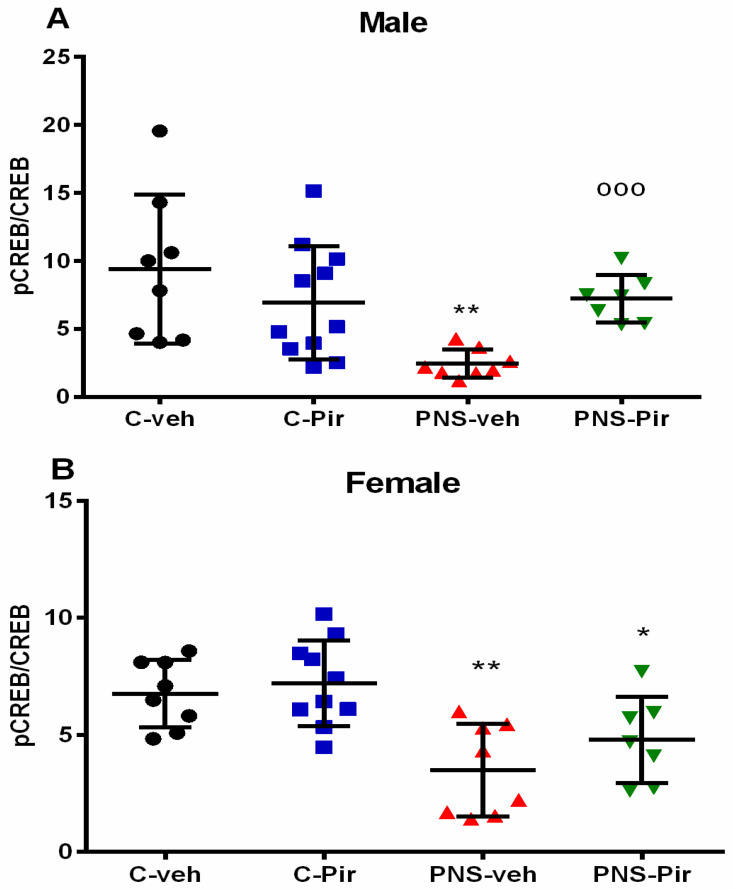
The chronic Piromelatine treatment exerted a sex-dependent elevation of pCREB/CREB ratio in the hippocampus in a PNS model. Three-way ANOVA indicated a main PNS effect [F_1,54_ = 14.482, *p* < 0.001], PNS × drug interaction [F_1,54_ = 6.943, *p* = 0.011], and interaction between PNS × sex × drug [F_1,54_ = 4.364, *p* = 0.041] in (**A**) male and (**B**) female offspring. Data are presented as means ± SEM: * *p* < 0.005 vs. C-veh, ^o^ *p* < 0.005 vs. PNS-veh, # *p* < 0.05 vs. male.

**Table 1 ijms-24-01271-t001:** The chronic Piromelatine treatment tended to increase melatonin levels in PNS male rats but did not affect the hormonal level in plasma of female rats with PNS (*p* > 0.05).

Group	Median ± SD	Range
Male		
C-veh	0.14 ± 0.09	(0.075 ± 0.24)
PNS-veh	0.058 ± 0.038	(0.05 ± 0.14)
PNS-Pir	0.85 ± 0.44	(0.092 ± 1.608)
Female		
C-veh	0.12 ± 0.79	(0.1–4.0)
PNS-veh	0.096 ± 0.083	(0.06 ± 0.42)
PNS-Pir	0.095 ± 0.03	(0.06 ± 1.6)

## Data Availability

Not applicable.

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
