# Peer review of "Sex-Dependent Effect of Chronic Piromelatine Treatment on Prenatal Stress-Induced Memory Deficits in Rats"

_ijms, 2023, doi:10.3390/ijms24021271_

Round 1

Reviewer 1 Report

This MS shows for the effects of chronic piromelatine in memory performance in PNS rats.  

In the object recognition test (ORT), please specify the interval between training and test session. In the figure 1, interval described is 120 min, while in the methods, authors declared 1 hour.

Control male rats in the ORT, discrimination index almost reached 1.0. How could the authors discard a biased preference for the novel object? Did the authors evaluated the natural preference for novel and familiar objects in order to avoid the use of preferred objects.

Please present the total time spent exploring both objects in the training and test sessions in the ORT. This information is relevant for ruling out possible effects of PNS or piromelatine on locomotion.

In the discussion, please specify why the mechanism of action of piromelatine is complex.

I suggest transferring the chemical name of piromelatine displayed in the discussion to the methods session.

How authors explain the discrepancy between ORT and RAM tests in PNS females?

Did the authors assess estrous cycle in females before behavioral tests? This information should be stated in the MS. Additionally, authors should consider the effects of estrus cycle in the memory of females.  

Some aspects should be clarified in the methods, in order to deeply understand how experiments were performed.   

Which day time were behavioral tests performed?

Regarding PNS, how many animals from each litter were used in each behavioral test? This question is relevant since each stressed mother is one subject, thus it is not fair to use a whole litter to compose the mean of a behavioral or biochemical data.

Were the same animals exposed to all behavioral tests? Is yes, which interval between tests were adopted?

Please specify in details random stressors used in the PNS. The sequence of stressors, duration, day time, etc, should be described in details.

In the ORT, please revise the formula applied to calculate discrimination index. Authors mention that data are multiplied by 100, but if yes, DI from controls would be higher than 50.

Which time blood to measure melatonin was collected?

Please revise the description of statistical analysis. Authors mention that they used three-way ANOVA to evaluate biochemical data, and they also mentioned that they used Mann-Whitney U-test as post-hoc test, but it did not happen.

Please revise throughout the manuscript: abbreviations must be written in full when first presented.

Author Response

Review #1

Dear Editor and Reviewers,

Thank you for the careful evaluation of our manuscript entitled “Sex-dependent effect of chronic Piromelatine treatment on prenatalstress-induced memory deficits in rats”. We have revised the manuscript taking into account the suggested modifications. All changes in the MS are highlighted.

Point #1. In the object recognition test (ORT), please specify the interval between training and test session. In the figure 1, interval described is 120 min, while in the methods, authors declared 1 hour.

Response: We are thankful for this comment. The method was updated accordingly.

Point #2. Control male rats in the ORT, discrimination index almost reached 1.0. How could the authors discard a biased preference for the novel object? Did the authors evaluated the natural preference for novel and familiar objects in order to avoid the use of preferred objects.

Response: Thank you for this question. The rats, which showed preference to one of the identical objects during the training session were excluded from the test. This explanation was included in the method in point 4.4.1.

Point #3. Please present the total time spent exploring both objects in the training and test sessions in the ORT. This information is relevant for ruling out possible effects of PNS or piromelatine on locomotion.

Response: We used the index, which is a reliable parameter for detection of cognitive reaction and to eliminate the locomotion. The rats, which showed preference to one of the identical objects during the training session were excluded from the statistics. There was no difference between the total time for sniffing in the training and the test sessions for the familiar objects, but these data are not shown. In previous studies we used more sensitive test to evaluate locomotion.

Point #4. In the discussion, please specify why the mechanism of action of piromelatine is complex.

Response: This information is explained in the introduction, where a description is present that Piromelatine is a multimodal drug with a complex mechanism of action, which simultaneously targets the melatoninergic and in part the serotoninergic systems by activating the MT type 1, 2 and 3 receptors and the serotonin (5-HT) type 1A and 1D receptors.

Point #5. I suggest transferring the chemical name of piromelatine displayed in the discussion to the methods session.

Response: It was moved as recommended.  

Point #6. How authors explain the discrepancy between ORT and RAM tests in PNS females?

Response: Both tests are intended to test different parameters: associative memory was detected by the ORT, while spatial learning and memory were evaluated by the RAM test and this difference is explained in the methods pints 4.4.1 and 4.4.2.

As we have explained in the discussion, in our previous study we have reported that BDNF expression was decreased in male offspring with PNS but was not affected in female offspring, suggesting that this signaling molecule in the hippocampus is critical for improved response in the RAM task of female rats with PNS. Please, also see response to point #7.

Point #7. Did the authors assess estrous cycle in females before behavioral tests? This information should be stated in the MS. Additionally, authors should consider the effects of estrus cycle in the memory of females.  

Response: We cannot exclude a possible effect of the estrous cycle on spatial memory in female rats. This is a limitation, which we are not able to avoid, because estrous cycle is very short (4-5 days) and RAM test is a chronic test starting with 1 week diet and then the test is conducted for another week. This was included in the text: page 9.

Point #8. Some aspects should be clarified in the methods, in order to deeply understand how experiments were performed.

Response: The method were updated.   

Point #9. Which day time were behavioral tests performed?

Response: The behavioral tests started from P74 (after the 13th injection of veh/Pir) and were conducted between 10:00 a.m. and 2:00 p.m. It was added in the text in point 4.4.

Point # 10. Regarding PNS, how many animals from each litter were used in each behavioral test? This question is relevant since each stressed mother is one subject, thus it is not fair to use a whole litter to compose the mean of a behavioral or biochemical data.

Response: Offsprings of at least 3 dams were used per group. It was added in the text in point 4.2.

Point #11. Were the same animals exposed to all behavioral tests? Is yes, which interval between tests were adopted?

Response: The same animals were used. The ORT was conducted at P74 (the pre-test) and P75 (the test).  After the ORT the rats were put on a diet for 1 week to be prepared for the one-week RAM test (P83-P89). It was added in the text in point 4.4.

Point #12. Please specify in details random stressors used in the PNS. The sequence of stressors, duration, day time, etc, should be described in details.

Response: The stressors were described in a previous article, which is cited in the method. The stressors were randomly applied, as it is stated in the method, to avoid habituation.

Point #13. In the ORT, please revise the formula applied to calculate discrimination index. Authors mention that data are multiplied by 100, but if yes, DI from controls would be higher than 50.

Response: Yes, this is correct, the DI from controls is higher than 50.

Point #14. Which time blood to measure melatonin was collected?

Response: The rats were sacrificed 3 days after the last behavioral test (RAM) on P92 2 hours be-fore the onset of the dark phase (4 p.m.). It was added in the text.

Point #15. Please revise the description of statistical analysis. Authors mention that they used three-way ANOVA to evaluate biochemical data, and they also mentioned that they used Mann-Whitney U-test as post-hoc test, but it did not happen.

Response: Thank you for this remark. This was corrected and three-way was done for all tests.

Point #16. Please revise throughout the manuscript: abbreviations must be written in full when first presented.

Response: Thank you, revised.

Reviewer 2 Report

Several comments have been included in the manuscript. Kindly correct.

Author Response

There is a link between melatonin,  the melatoninergis system and CREB, as well as CREB can be activated via  5-HT. This drug targets the melatonin MT1/MT2 receptors and 5-HT type 1A receptors, which are widely expressed in brain regions involved in learning and memory processes. The introduction has been updated,There is a link between melatonin,  the melatoninergis system and CREB, as well as CREB can be activated via  5-HT. This drug targets the melatonin MT1/MT2 receptors and 5-HT type 1A receptors, which are widely expressed in brain regions involved in learning and memory processes. The introduction has been updated,There is a link between melatonin,  the melatoninergis system and CREB, as well as CREB can be activated via  5-HT. This drug targets the melatonin MT1/MT2 receptors and 5-HT type 1A receptors, which are widely expressed in brain regions involved in learning and memory processes. The introduction has been updated,There is a link between melatonin,  the melatoninergis system and CREB, as well as CREB can be activated via  5-HT. This drug targets the melatonin MT1/MT2 receptors and 5-HT type 1A receptors, which are widely expressed in brain regions involved in learning and memory processes. The introduction has been updated,There is a link between melatonin,  the melatoninergis system and CREB, as well as CREB can be activated via  5-HT. This drug targets the melatonin MT1/MT2 receptors and 5-HT type 1A receptors, which are widely expressed in brain regions involved in learning and memory processes. The introduction has been updated,There is a link between melatonin,  the melatoninergis system and CREB, as well as CREB can be activated via  5-HT. This drug targets the melatonin MT1/MT2 receptors and 5-HT type 1A receptors, which are widely expressed in brain regions involved in learning and memory processes. The introduction has been updated,
